# The Ability of Chlorophyll to Trap Carcinogen Aflatoxin B_1_: A Theoretical Approach

**DOI:** 10.3390/ijms23116068

**Published:** 2022-05-28

**Authors:** Alma Vázquez-Durán, Guillermo Téllez-Isaías, Maricarmen Hernández-Rodríguez, René Miranda Ruvalcaba, Joel Martínez, María Inés Nicolás-Vázquez, Juan Manuel Aceves-Hernández, Abraham Méndez-Albores

**Affiliations:** 1Unidad de Investigación Multidisciplinaria L14 (Alimentos, Micotoxinas, y Micotoxicosis), Facultad de Estudios Superiores Cuautitlán, Universidad Nacional Autónoma de México, Cuautitlán Izcalli, Estado de Mexico 54714, Mexico; almavazquez@comunidad.unam.mx (A.V.-D.); juanmanuel.is.acevesh@gmail.com (J.M.A.-H.); albores@unam.mx (A.M.-A.); 2Department of Poultry Science, University of Arkansas, Fayetteville, AR 72701, USA; gtellez@uark.edu; 3Laboratorio de Cultivo Celular, Sección de Posgrado e Investigación, Escuela Superior de Medicina, Instituto Politécnico Nacional, Ciudad de Mexico 11340, Mexico; dra.hernandez.ipn@gmail.com; 4Departamento de Ciencias Químicas, Facultad de Estudios Superiores Cuautitlán Campo 1, Universidad Nacional Autónoma de México, Avenida 1o de Mayo s/n, Colonia Santa María las Torres, Cuautitlán Izcalli, Estado de Mexico 54740, Mexico; mirruv@yahoo.com.mx; 5Facultad de Ciencias Químicas, Universidad Autónoma de San Luis Potosí, San Luis Potosi 78210, Mexico

**Keywords:** chlorophyll a, aflatoxin B_1_, intermolecular interactions, molecular modeling, density functional theory, M06-2X functional

## Abstract

The coordination of one and two aflatoxin B_1_ (AFB_1_, a potent carcinogen) molecules with chlorophyll a (**chl a**) was studied at a theoretical level. Calculations were performed using the M06-2X method in conjunction with the 6-311G(d,p) basis set, in both gas and water phases. The molecular electrostatic potential map shows the chemical activity of various sites of the AFB_1_ and **chl a** molecules. The energy difference between molecular orbitals of AFB_1_ and **chl a** allowed for the establishment of an intermolecular interaction. A charge transfer from AFB_1_ to the central cation of **chl a** was shown. The energies of the optimized structures for **chl a** show two configurations, unfolded and folded, with a difference of 15.41 kcal/mol. **Chl a** appeared axially coordinated to the plane (α-down or β-up) of the porphyrin moiety, either with the oxygen atom of the ketonic group, or with the oxygen atom of the lactone moiety of AFB_1_. The complexes of maximum stability were **chl a 1**-α-E-AFB_1_ and **chl a 2**-β-E-AFB_1_, at −36.4 and −39.2 kcal/mol, respectively. Additionally, with two AFB_1_ molecules were **chl a 1**-D-2AFB_1_ and **chl a 2**-E-2AFB_1_, at −60.0 and −64.8 kcal/mol, respectively. Finally, biosorbents containing chlorophyll could improve AFB_1_ adsorption.

## 1. Introduction

Several fungi, such as *Aspergillus flavus*, *A. parasiticus*, *A. nomius*, and *A. psudotamarii*, can produce toxic secondary metabolites recognized as mycotoxins [1,2]. Among the toxins produced by these fungi, aflatoxin B_1_ (AFB_1_) (Figure 1a) exhibits the most potent mutagenic and carcinogenic potential. Consequently, it has been categorized as a human Group 1 carcinogen by the International Agency of Research on Cancer [3]. In addition, several outbreaks of human aflatoxicosis have been reported in emergent countries [4,5,6,7]. Therefore, investigation of dietary agents that selectively bind AFB_1_ in the gastrointestinal tract and reduce their bioavailability must be considered essential.

Chlorophylls are among the most abundant pigments in nature; this class of compounds contains four linked pyrrole rings and a hydrophobic side chain of phytol. Chlorophyll a (**chl a**) (Figure 1b) and chlorophyll b (**chl b**) are the two most significant chlorophylls present in higher plants, green algae, and some prochlorophytes. These molecules differ in that there is an aldehyde group in **chl b** instead of a methyl group, present in **chl a**, located at the pyrrole ring N3. It is important to note that many studies have shown that chlorophylls have significant anticarcinogenic activity against a wide range of potential human carcinogens, including AFB_1_ [8,9,10,11]. Therefore, different mechanisms responsible for cancer-preventative activity have been proposed, including antioxidant activity [12,13], modulation of detoxification pathways [14], induction of apoptosis [15], and carcinogen trapping [16,17,18,19,20]. However, the mechanisms involved in the anticarcinogenic action of chlorophylls and their potential for human chemoprevention against AFB_1_ are poorly understood. Furthermore, chlorophyllin (**chl**), a water-soluble derivative of chlorophyll, could also create strong non-covalent complexes with carcinogens. Using molecular modeling and experimental studies, it has been determined that **chl**–AFB_1_ interactions are extremely energetically favorable (up to −20 kcal/mol), involving both electrostatic attractions and van der Waals interactions [21,22].

As a part of our research, we have previously performed important studies contributing to the understanding of AFB_1_, summarized as follows: (a) A mass spectrometry/mass spectrometry study on the degradation of AFB_1_ in maize with aqueous citric acid [23]. (b) In addition to several theoretical calculations, applying density functional theory, confirming that the active site corresponds to the carbonylic functionality of the lactonic moiety [24]. The performance of quantum mechanical calculations to explain the chemical behavior of the lactone ring of AFB_1_ hydrolyzed under acidic conditions, suggesting the deletion of its carcinogenic properties. (c) A theoretical study [25] of 8-chloro-9-hydroxy-AFB_1_, carried out by DFT, determining the structural, electronic, and spectroscopic properties of this reaction product of AFB_1_, allowing for its theoretical characterization. (d) A theoretical study [26] related to the adsorption process of B-aflatoxins using a vegetable specimen *Pyracantha koidzumii* (Hayata); the interaction of AFB_1_ with the functional groups present in the biosorbent was investigated. (e) In a recent publication [27], several in vitro experiments were conducted to evaluate the effectiveness of lettuce and field horsetail as biosorbents for the removal of aflatoxin AFB_1_. In conclusion, several physicochemical interactions with chlorophylls are involved in the adsorption process [17,22,28,29].

Considering the previously disclosed information, the goal of this work is to provide novel knowledge related to the interaction between AFB_1_ and **chl a** using theoretical calculations employing density functional theory (DFT). In other words, appropriate interactions between AFB_1_ and **chl a** were achieved. As a result, a careful in silico study was performed using geometry optimization, molecular electrostatic potential maps, and the highest occupied molecular orbital–lowest unoccupied molecular orbital gap, explaining the progress of several **chl a**–AFB_1_ complexes.

## 2. Results

### 2.1. DFT Optimized Structures

#### Determination of Most Stable Conformer between **chl a 1** and **chl a 2**

Figure 1 shows the chemical structures of AFB_1_ and **chl a** with appropriate nomenclature; this will make this section easier to comprehend. The energies of the corresponding optimized structures for **chl a** show two different configurations, as shown in Figure 2. In the first instance, there is a phytol moiety on the N1 ring in an unfolded form (**chl a 1**) [30,31] with −2934.13 Hartrees. On the other hand, there is a phytol chain with a folded structure (**chl a 2**) and −2934.15 Hartrees, meaning that conformer **chl a 2** is the most stable, with a difference of 15.41 kcal/mol (Figure 2). This, may be due to weak hydrogen bond interactions [32] between the lone pair of electrons at the nitrogen atoms and nearby hydrogens atoms of the phytol moiety, resulting in the following data: C5H2–N4: 2.86 Å, C11H–N3: 2.9 Å, and C14H2–N2: 2.63 Å. After searching in the literature, we found that Alvarado-González et al. [30] and Schulte et al. [31] reported this more stable conformation (**chl a 2**). In preceding works by Kobayashi and Reimers [33] and Chen and Cai [34], it was stated that if a methyl group replaces the phytol chain, non-significant consequences can be expected; however, in this work, the bulky chain plays an important role in the stereo structure adopted by the chlorophyll. Therefore, this moiety should not be underestimated.

AFB_1_, the most stable conformer, displayed an energy value of −1106.22 Hartrees. In general, the molecule is planar, but with the A ring perpendicular to the plane.

In general, the optimized structure calculations were established in the gas phase.

### 2.2. Structural and Energetic Parameters from Optimized Conformers

#### 2.2.1. Atomic Charges of **chl a 1**, **chl a 2**, AFB_1_, and Their Complexes

Atomic charge is not a physically obvious parameter; its value depends on the scheme by which the electron density of a molecule is partitioned. Additionally, a natural charge results from natural population analysis (NPA), converting a wave function’s molecular orbitals from delocalized to localized, maintaining all information contained in the wave function [35]. Moreover, NPA has been recommended to overcome problems associated with other schemes of charge [36].

Related to the above commentary, Appendix A displays the atomic charges of **chl a 1**, **chl a 2**, and their corresponding complexes, considering one and two molecules of AFB_1_. Accordingly, Appendix A displays the atomic charges of AFB_1_. Theoretical calculations showed that the most negative charge in the complexes is generally localized on N1 (−0.737–796 e^−^), except for **chl a 1**-α-E-AFB_1_ (−0.698 e^−^). However, for the uncoordinated **chl a 1** and **chl a 2**, higher values were obtained: −0.792 and −0.796 e^–^, respectively.

The nitrogen atoms, N2, N3, and N4, for **chl a 1** and **2**, exhibit similar charge values: For the **chl a 1** complex with one AFB_1_, $\bar{X}=$ −0.724 e^−^ for N2, $\bar{X}=$ −0.731 e^−^ for N3, and $\bar{X}=$ −0.703 e^−^ for N4; for the **chl a 2** complex with one AFB_1_, $\bar{X}=$ −0.719 e^−^ for N2, $\bar{X}=$ −0.745 e^−^ for N3, and$\;\bar{X}=$ −0.695 e^−^ for N4. For **chl a 1** with two AFB_1_ molecules, the charges are $\bar{X}=$ −0.718 e^−^ for N2, $\bar{X}=$ −0.732 e^−^ for N3, and $\bar{X}=$ −0.667 e^−^ for N4. For **chl a 2** with two AFB_1_ molecules, the charges are $\bar{X}=$ −0.698 e^−^ for N2, $\bar{X}=$ −0.7185 e^−^ for N3, and $\bar{X}=$ −0.671 e^−^ for N4.

Concerning AFB_1_, the most negative charge is located on the oxygen atoms of the carbonyl groups, showing similar charge values of −0.534 and −0.533 e^–^. Meanwhile, the major hydrogen acidity atoms corresponded to H9a > H9 > H8 > H6a, 0.242, 0.236, 0.201, and 0.183 e^–^, respectively.

On the other hand, for all the target molecules, the site with higher positive charge is located on the Mg^2+^ ion (1.749–1.766 e^−^). In particular, the obtained data are as follows: **chl a 1**, 1.758 e^–^; **chl a 1** complexes with one AFB_1_, 1.757–1.760 e^−^; **chl a 1** complexes with two AFB_1_, 1.753 e^−^; **chl a 2**, 1.762 e^−^; **chl a 2** complexes with one AFB_1_ molecule, 1.760–1.766 e^−^; for **chl a 2** complexes with two AFB_1_, 1.749–1.756 e^−^.

#### 2.2.2. Bond Distance (Å) of Optimized Geometries

Table 1 shows the bond distances for **chl a 1**, **chl a 2**, and their complexes. The distances were taken between the magnesium ion and the oxygen or nitrogen atom.

Once the optimized conformers were established, and the charges of all atoms were acquired, the subsequent strategy was to recognize the reactive sites in the target molecules. Accordingly, in Appendix A, the obtained electrostatic potential maps for AFB_1_, **chl a 1**, and **chl a 2** are displayed; in the case of AFB_1_, the electrostatic potential [37] is more reactive (negative, red color) around the oxygen atoms localized on D and E rings, indicating that these positions may perform electrophilic attacks, and highlighting that, as in previous studies, the D ring (lactone moiety) is the active site [24].

Regarding the chlorophyll conformers, the light red color of the oxygen atoms of both the ketonic and the ester carbonyl groups positioned on the N4 ring is indicative of a negative area. An interesting zone around the magnesium ion (positive, blue color) reveals that this position may be subject to a nucleophilic attack [21,34]. Since the porphyrinic moiety is planar, the magnesium ion could display two axial nucleophilic interactions. Thus, the results are congruent with the data on atomic charges.

#### 2.2.3. Molecular Orbitals: HOMO–LUMO for **chl a 1**, **chl a 2**, and AFB_1_

The highest occupied, and the lowest unoccupied, molecular orbitals were determined to reveal the stability of the studied molecules, as show in Appendix A. Only the HOMO for AFB_1_ and the LUMO for **chl a 1** and **chl a 2** were considered due to their corresponding nucleophilic and electrophilic character. The HOMO for AFB_1_, −7.604 eV, is mainly located at the oxygen atoms on rings D and E, with significant contributions by the aromatic system ring C. Thus, these rings must be considered as the interaction sites when AFB_1_ acts as an electron-donor specie, interacting with a cationic center [38] (Appendix A). The LUMO for **chl a 2**, −2.295 eV, and **chl a 1**, −2.126 eV, is located on the double bonds of the porphyrin system (π bonds and lone-electron-pair N atoms), according to Alvarado-González et al. [30] and Bechaieb et al. [39]. The energy gap is smaller between the HOMO of AFB_1_ and LUMO of **chl a 2**; therefore, they could accomplish a better interaction than **chl a 1** with the HOMO of AFB_1_ (Appendix A). This assumption is made since the gap (E_LUMO_ − E_HOMO_) in the interaction of **chl a 2**–AFB_1_ has a value of 5.309 eV, and the gap for **chl a 1**–AFB_1_ shows a value of 5.477 eV, confirming that the **chl a 2**–AFB_1_ complex is the most stable. Considering the maximum hardness principle [40,41], which asserts that the systems tend to be hard when they show significant gap energy, the AFB_1_ molecule is hard, showing a gap of 6.324 eV. **Chl a 1** and **chl a 2** have gap values of 3.799 and 3.824 eV, respectively. Thus, the chlorophyll molecule is soft. Nevertheless, the interaction between AFB_1_–HOMO and **chl a**–LUMO is favored.

Finally, the frontier electron density can be used to predict the most reactive position in a π electron system. Hence, considering, that HOMO–LUMO and its property (energy) is very useful to estimate the chemical reactivity of molecules, the interactions between the vacant orbital (LUMO of **chl a 2**) and the electron pair (HOMO of AFB_1_) displayed small energy separation. It can also be assumed that with the overlapping between the orbitals of two target molecules, the stability is improved. In addition, as shown in Appendix A, the site and sharpness of some lobes display noticeable changes, but not intensely.

### 2.3. Geometry of Unfolded and Folded Chlorophyll with One AFB_1_ Molecule

#### 2.3.1. Frontal View of Unfolded and Folded Chlorophyll

Unfolded (Figure 3a) corresponds to the phytol chain in an extended form. In contrast, folded (Figure 3b) corresponds to the phytol chain positioned under the plane of a porphyrin moiety. In addition, Figure 3 shows the axial possibilities of coordination, α implies AFB_1_ placed under the chain and, consequently, β is positioned on the up side [42].

#### 2.3.2. Geometry of Unfolded Chlorophyll with One AFB_1_ Molecule

In the computed ground state of **chl a**, the Mg^2+^ ion could be axially coordinated by β and α to the oxygen atom of the ketone group, ring E, or with the oxygen atom of the carbonylic lactone group, ring D, such that C1=O12 or C11=O14, respectively. Therefore, it was considered appropriate to carry out these axial interactions, employing both **chl a 1** and **chl a 2**, conformers, as shown in Appendix A. These interactions were proposed since Mg^2+^ yields more stable complexes with oxygen atom donors than with nitrogen atom donors, and considering that the coordination number of Mg^2+^ in porphyrins could be five or six [43,44].

The first offered interaction of **chl a 1** is with the oxygen molecule (C=O) on ring D, and the Mg–O distance in the optimized geometry is 2.101 Å, as shown in Appendix A and Table 1. The distance to the oxygen (C=O) on ring E is 2.106 Å, as shown in Appendix A and Table 1. Similar data were reported by Heimdal et al. (2.18 Å) [45], Rutkowska et al. (2.19 Å) [43], and Timkovich et al. (2.10 Å) [46] for complexes of Mg^2+^–H_2_O, and 1.9–2.3 Å for Mg–RNA complexes [47]. The smaller value for **chl a 1**-α-D–AFB_1_ is explained considering a minor steric hindrance. This effect plays an important role, because in the **chl a 1**-α-D–AFB_1_ complex, the C11=O14 of AFB_1_ is α-located, with a small section between N1 and N4, as shown in Figure 1 and Appendix A. Related to C1=O12, in the **chl a 1**-α-E–AFB_1_ complex, the AFB_1_ is α-positioned, again with a small section between N1 and N4, as shown in Figure 1 and Appendix A. This confirms that the D ring (lactone moiety) is the most active site, which is in agreement with data from the literature [24].

On the other hand, in the **chl a 1**-β-D–AFB_1_ complex, the Mg–O distance has a value of 2.124 Å, as shown in Appendix A and Table 1. For the **chl a 1**-β-E–AFB_1_ complex, this value is 2.111 Å, as show in Appendix A and Table 1.

In the complex with the higher-energy value (**chl a 1**–AFB_1_), the Mg–N interatomic distances are longer in comparison to uncoordinated **chl a 1** (see Table 1). This behavior must be due to a saturated bond in the N1 ring. It has been reported that the Mg–N bond can be elongated to 0.02–0.04 Å by neutral ligands [45]. In this work, the resulting distance order for all complexes was N1 > N3 > N2 > N4, in agreement with previous reports [48,49,50]. The Mg^2+^–AFB_1_ complex caused small but noticeable changes in the geometry of the porphyrin moiety of **chl a 1** due to a change in the coordination number (4 to 5) of the Mg^2+^ ion. It is important to note that in **chl a 1**-α-D–AFB_1_, the substituents on N2 (vinyl) and N3 (methyl) change their configuration. Additionally, for **chl a 1**-α-E–AFB_1_, **chl a 1**-β-D–AFB_1_, and **chl a 1**-β-E–AFB_1_, a change in configuration is also perceived for the vinyl group on N2.

In general, the Mg^2+^ ion coordinated with the nitrogen atoms yielded a square-based pyramid geometry with the Mg^2+^ located out-of-plane due to the coordination number (5), in agreement with literature data [35,47,48,49,50,51,52,53,54,55,56]. A displacement of the Mg^2+^ ion toward the axial ligand was perceived from 0.57 to 0.68 Å; according to Heimdal et al. [45] and Zucchelli et al., this displacement was 0.43–0.54 Å [52]. In addition, AFB_1_ caused a relatively strong polarization of the chlorophyll molecule [54,57,58].

#### 2.3.3. Geometry of Unfolded Chlorophyll with Two AFB_1_ Molecules

Since six is the common coordination number of the magnesium ion [43] and satisfies its coordination sphere, it was proposed as the coordination number of chlorophyll (**chl a 1** and **chl a 2**) with two AFB_1_ molecules. Accordingly, the β and α coordinations with C1=O12 or C11=O14 (rings E and D) were also considered, as shown in Appendix A, respectively. Additionally, the magnesium atom exhibits a coordination number of five, the water molecule being the fifth axial ligand [44,55].

Thus, for the interaction between oxygen in ring E and **chl a 1** (**chl a 1**-E–2AFB_1_ complex), the corresponding Mg–O distances for β and α interactions were 2.228 and 2.351 Å, as shown in Appendix A and Table 1. For the analog interactions with the oxygen in ring D (**chl a 1**-D–2AFB_1_ complex), the Mg–O distances were 2.230 and 2.428 Å for β and α interactions, respectively, as shown in Appendix A and Table 1. These values agree with those reported by Ben Fredj et al. (2.24 Å) [44], Rutkowska et al. (2.26 Å) [43], and the values between 2.210 and 2.267 Å reported by Ghosh et al. [56].

The previously mentioned values are smaller in both complexes due to steric hindrance with N2 and N3 rings. Hence, it is convenient to highlight that, for the **chl a 1**-D–2AFB_1_ complex, the AFB_1_ molecule is more internally situated, in comparison to the **chl a 1**-E–2AFB_1_ complex, generating more steric hindrance and consequently larger bond distances.

Regarding the Mg–N interatomic distances, they were longer in both previously mentioned complexes in comparison with that in the uncoordinated **chl a 1**. This is shown in Table 1. Moreover, the previous order established for the distances of Mg–N, N1 > N3 > N2 > N4, is also maintained in these complexes [48,49,50]. Consequently, the Mg^2+^–2AFB_1_ complex achieved small but noticeable changes in the configuration of the vinylic group on N2. It is important to highlight that the Mg^2+^ ion satisfies their coordination sphere, displaying an octahedral geometry due to the acquired coordination number of six, setting the magnesium ion in a plane [51,52,53,56,59]. Thus, the Mg–N bond distances decreased [44].

#### 2.3.4. Geometry of Folded Chlorophyll with One AFB_1_ Molecule

For the **chl a 2**–AFB_1_ complex, axial attacks on the Mg^2+^ ion by oxygen atoms (C1=O12 or C11=O14) were considered. The first displayed interaction was with the oxygen on ring D; the Mg–O distance in the optimized geometry was 2.123 Å (Appendix A, Table 1). The distance to the oxygen on ring E is 2.080 Å (Appendix A, Table 1); similar data were reported by Heimdal et al. (2.18 Å) [45], Rutkowska et al. (2.19 Å) [43], and Timkovich et al. (2.10 Å) [46]. The minor value for **chl a 2**-α-E–AFB_1_ is due to a steric hindrance, since the AFB_1_ molecule is located under N1 and N4. For the **chl a 2**-α-D–AFB_1_ complex, the AFB_1_ molecule is under N1, and ring A of AFB_1_ is located under N3.

On the other hand, for the **chl a 2**-β-E–AFB_1_ complex, the Mg–O distance has a value of 2.120 Å (Appendix A, Table 1), and that for the **chl a 2**-β-D–AFB_1_ complex was 2.119 Å (Appendix A, Table 1). Again, the most active site is the D ring [24].

Moreover, in the higher-energy **chl a 2**–AFB_1_ complex, the Mg–N interatomic distances are longer than in uncoordinated **chl a 2** (Table 1) due to the saturated bond on the N1 ring, as previously reported (0.02 to 0.04 Å) [45]. In this work, the resulting distance order for all complexes was N1 > N3 > N2 > N4, in agreement with previous reports [48,49,50]. Additionally, the Mg^2+^–AFB_1_ complex caused small but noticeable changes in the geometry of the porphyrin moiety of **chl a 2**-α-D–AFB_1_, **chl a 2**-α-E–AFB_1_, and **chl a 2**-β-E–AFB_1_, which was due to a change in the coordination number (four to five) of the Mg^2+^ ion. As in the case of **chl a 1**, comparable changes in configuration for the substituents were perceived, and for **chl a 2**-β-D–AFB_1_, a change was observed in the methyl substituent on N3. As previously shown for **chl a 1**, the complexes also yielded a square-based pyramid geometry [35,47,48,49,50,51,52,53,54,55,56]. A displacement of the Mg^2+^ ion toward the axial ligand was perceived from 0.57 to 0.68 Å [45]; according to Zucchelli et al., this displacement was 0.43–0.54 Å [52]. Additionally, the AFB_1_ caused a comparatively robust polarization in the chlorophyll molecule [54,57,58].

#### 2.3.5. Geometry of Folded Chlorophyll with Two AFB_1_ Molecules

For the β and α interactions with the oxygen on ring E of the **chl a 2**-E–2AFB_1_ complex, the Mg–O distances were 2.251 and 2.186 Å, respectively (Appendix A, Table 1). In addition, for the interactions with the oxygen on ring D of the **chl a 2**-D–2AFB_1_ complex, the corresponding Mg–O distances were 2.285 and 2.167 Å (Appendix A, Table 1). These values are in agreement with those reported by Ben Fredj et al. (2.24 Å) [44], Rutkowska et al. (2.26 Å) [43], and the 2.210 to 2.267 Å reported by Ghosh et al. [56]. The α values are shorter in both complexes due to steric hindrance.

In addition, Mg–N interatomic distances are slightly longer than those of **chl a**, as shown in Table 1. This is due to the saturated bond on the N1 ring, which was 2.131 Å and 2.128 Å for **chl a** and complex **chl a 2**-D–2AFB_1_, respectively. The other Mg–N bonds exhibited elongation from 0.02 to 0.04 Å [45,46,47,48]; the previous order established for the distances of Mg–N, N1 > N3 > N2 > N4, is also maintained in this complex type [48,49,50]. Additionally, the Mg^2+^–AFB_1_ complex yielded changes in the configuration of substituents on N2; the Mg^2+^ ion satisfied its coordination sphere, showing an octahedral geometry [51,52,53,56,59], and the Mg–N bond distances decreased when the magnesium atom came into the plane [44].

### 2.4. Interaction Energy of the Complexes in Gas Phase

#### 2.4.1. Considering the Coupling of One AFB_1_ Molecule with **chl a 1** and **chl a 2**

In **chl a 1**, considering both α and β configurations, the **chl a 1**-α-E–AFB_1_ complex was more stable, at −36.4 kcal/mol, as shown in Appendix A, with differences between 0.7 and 4.3 kcal/mol in comparison with the other complexes. In view of the energy differences between the carbonyl groups in these four complexes, the observed energy values were 0.9–3.6 kcal/mol. When AFB_1_ is β-oriented, the energetic difference between C1=O12 and C11=O14 is not significant. However, if it is α-oriented, important energy differences are perceived due to steric hindrance in the phytol chain.

Regarding the four **chl a 2** complexes, **chl a 2**-β-E–AFB_1_, with a value of −39.2 kcal/mol, was more stable, as shown in Appendix A. It had a range of 2.7–6.4 kcal/mol in comparison to the other related complexes. In addition, a difference was detected between the carbonyl groups in the complexes. Thus, when the configuration of AFB_1_ is α, the difference is 2.7 kcal/mol. However, if AFB_1_ is β-oriented, the difference is 0.4 kcal/mol. In general, considering the eight interactions, **chl a 2**-β-E–AFB_1_, with a value of −39.2 kcal/mol, was the most stable of all.

#### 2.4.2. Considering the Coupling of Two AFB_1_ Molecules with **chl a 1** and **chl a 2**

Appendix A summarizes the energy of interaction values of the corresponding double coupling. Regarding the energy interactions between **chl a 1**-E–2AFB_1_ and **chl a 1**-D–2AFB_1_, the second was more stable, at −62.3 kcal/mol. The difference between them was 2.2 kcal/mol. Comparing **chl a 2**-E–2AFB_1_ and **chl a 2**-D–2AFB_1_ complexes, the first was more stable (−64.8 kcal/mol), with a difference between them of 6.7 kcal/mol.

Concerning the four double complexes, the **chl a 1** double coupling had energy value differences of 11.2 and 20.1 kcal/mol for D and E, respectively, compared to the **chl a 2** analogs.

### 2.5. Interaction Energy of the Complexes in Water as a Solvent

In general, in the presence of water (ε = 78.4) as a solvent, the observed values in Appendix A implicate that the complexes’ stabilities diminish. However, the same tendency is exhibited in both evaluations. For example, as can be seen in Appendix A, the **chl a 2**-β-E–AFB_1_ complex presents an interaction energy of −39.2 kcal/mol in the gas phase. The same molecule showed a value of −27.9 kcal/mol in the solvent (Appendix A); in other words, this complex is the most stable under both conditions (gas vs. solvent). The above may be because water exerts an effect as a coordination agent, causing the Mg^2+^ ion to adopt a different geometry to the square-based pyramid, or that water molecules solvate the AFB_1_ or **chl a** [51,60,61]. Regarding the **chl a**–H_2_O complex, Kobayashi and Reimers reported interaction energy values of −15.3, −14.9, and −15.1 kcal/mol, determined using MP2, DFT ωB97XD, and PBE-D3 methods, respectively [34]. According to the latter points and the obtained results, the solvent molecules could display an interaction with **chl a**.

On the other hand, double complexes also displayed more stability in the gas phase (−64.8 to −58.0 kcal/mol) compared to water (−49.0 to −45.9 kcal/mol). This could be because water solvates the AFB_1_ molecule, preserving the octahedral geometry of **chl a** [51,52,53,56,59]. The diaxial complex (**chl a**–2H_2_O) is unstable, in agreement with Fredj and Ruiz-López [60]; consequently, the water will not be able to displace the AFB_1_ molecules in the coordination with the magnesium ion.

### 2.6. Weak Hydrogen Bond Interactions between the Ester Functions of **chl a 1** and **chl a 2** with AFB_1_

All points made in the previous paragraphs are correlated, and mainly consider interactions with the magnesium ion. In this section, the occurrence of weak hydrogen bond interactions [33] between the ester functions of **chl a 1** and **chl a 2** with AFB_1_ are discussed and displayed in Appendix A and Figure 4a–d. It is important to remember that the numeration of several atoms is shown in Figure 1.

#### 2.6.1. **Chl a 1**–AFB_1_a (Three Hydrogen Bond Interactions)

The **chl a 1**–AFB_1_a complex (Figure 4a) shows three interactions: H8 of AFB_1_ and the oxygen of the ester carbonyl group of the phytol chain; O7 (furane) of AFB_1_ and the double α-hydrogen atom to both ketonic and ester groups in **chl a 1**; H6a of AFB_1_ with ketonic oxygen near N4. These have corresponding bond lengths of 2.237, 2.237, and 2.549 Å.

#### 2.6.2. **Chl a 1**–AFB_1_b (Three Hydrogen Bond Interactions)

As can be seen in Figure 4b, the **chl a 1**-AFB_1_b complex shows three weak hydrogen bonds: H6a atom of AFB_1_, proceeding as a bifurcated donor, has interactions with both oxygens of the ester groups of **chl a**, and the corresponding bond lengths are 2.506 Å and 2.595 Å. The carbonylic oxygen of the acetate group in the phytol chain acts as a bifurcated acceptor with H6a and H9a of the AFB_1_, exhibiting bond lengths of 2,343Å and 2,595 Å.

The **chl a 1**-AFB_1_a interactions are more stable than the **chl a 1**-AFB_1_b interactions, with a difference between them of 1.4 kcal/mol. Nevertheless, reverse behavior is shown in solvent conditions, at 1.1 kcal/mol.

#### 2.6.3. **Chl a 2**–AFB_1_c (One Hydrogen Bond Interaction)

**Chl a 2**–AFB_1_c (Figure 4c) displays only one weak hydrogen bond between hydrogen H6a of AFB_1_ and the phytol acetate’s carbonylic oxygen; the resulting bond length is 2.444 Å.

#### 2.6.4. **Chl a 2**–AFB_1_d (Two Hydrogen Bond Interactions)

Figure 4d presents two weak hydrogen interactions for **chl a 2**–AFB_1_, between hydrogen atoms H6a and H9a and the carbonylic oxygen of the phytol acetate. This oxygen atom acts as a bifurcated acceptor, displaying bond lengths of 2.385 Å and 2.436 Å. The **chl a 2**–AFB_1_c interactions are more stable than the **chl a 1**–AFB_1_d interactions, with a difference between them of 9.2 kcal/mol in the gas phase and 8.6 kcal/mol under solvent conditions. These four complexes show interaction energies lower than the previous complexes. In addition, the **chl a 2**–AFB_1_ interactions are more stable than in **chl a 1**–AFB_1_. The intermolecular hydrogen bonding interactions are less strong with the solvent effect.

The **chl a 2**–AFB_1_c interactions are more stable than in **chl a 1**–AFB_1_d. The differences between them are 9.2 kcal/mol in the gas phase and 8.6 kcal/mol under solvent conditions.

### 2.7. Docking Studies for **chl a 1**– and **chl a 2**–AFB_1_ Complexes

The docking studies were performed employing **chl a 1** and **chl a 2** as receptors and AFB_1_ as a ligand. Figure 5 shows that the binding mode between **chl a 2** and AFB_1_ show similar behavior to that previously discussed. **Chl a 2**– and **chl a 1**–AFB_1_ complexes display similar ΔG values, at −6.99 and −6.61 kcal/mol, respectively. Thus, as shown in Figure 5a for **chl a 2**, the Mg^2+^ ion interacts with the lactonic group (ring D) of the AFB_1_ molecule, revealing Mg–O distances with values of 5.2 and 3.9 Å, respectively. For **chl a 1**, as shown in Figure 5b, the Mg^2+^ ion interacts with the oxygen of the methoxy group on ring C, with an Mg–O distance of 4.4 Å. These results implicate that the conformation of **chl a 2** is preferred (−2934.15 Hartrees). This was reinforced with the HOMO–LUMO data, in agreement with the smaller energy gap.

### 2.8. Molecular Dynamics (MD) Simulations for **chl a 2**

MD simulations employ Newton’s law to evaluate the motions of complex systems, thus reproducing the behavior of the biological environment, including water molecules. Specifically, they show the changes in conformational states, which are very important in understanding the recognition pattern of macromolecule–ligand complexes. In this sense, MD simulations are used to identify the motions that can be modeled using this methodology [62]. To perform MD simulations, **chl a 2** was selected as a starting molecule because it is the most stable conformation (−2934.15 Hartrees) in comparison with unfolded conformer **chl a 1** [31,32].

Appendix A shows the root-mean-square deviation (RMSD) and radius of gyration of **chl a 2** obtained from 50 ns of MD simulations. The RMSD values represent the stability of the simulation and allow for the estimation of the equilibration timescale of the simulation. Thus, in Appendix A, it is possible observe that RMSD varies from 2.1 to 6.6 without reaching confluence, indicating that **chl a 2** presents conformational changes along the entire simulation [63]. In addition, it is possible to perceive a variability in radius of gyration values (Appendix A). The radius of gyration is a measure of the compactness of a ligand; in this case, the values can be deduced in folding and unfolding processes [64]. As can be seen, these changes in values are representative of folding and unfolding processes of **chl a 2**.

Figure 6 shows snapshots of every 10 ns of MD simulation of **chl a 2**. As can be seen, **chl a 2** displays a conformational change during the entire simulation. Interestingly, at 40 ns of simulation, **chl a 2** be disposed to acquire an unfolded conformation, and at 50 ns, **chl a 2** begins to returns to its folded conformation. Thus, with this result, it is possible to propose and verify that **chl a 2**, the most stable conformer, can exist in both conformational states, changing from folded to unfolded. Consequently, with a longer simulation, **chl a 2** could adopt its initial stable conformation.

### 2.9. Correlation between Experimental and Theoretical Findings

According to previous results [27], the biosorbents (lettuce, field horsetail, and Formosa firethorn) [26] tested in an in vitro study displayed excellent biosorption of AFB_1_, employing 0.1% *w*/*v* doses (95, 71, and 60% for lettuce, field horsetail, and Formosa firethorn, respectively). It is important to highlight the case of the lettuce, which showed a higher quantity of **chl** (5.10 mg/g sorbent) in comparison to field horsetail and Formosa firethorn (1.32 and 3.14 mg/g sorbent, respectively). Moreover, it is worth noting that lettuce possesses more **chl a** (from 1.6- to 3.8-fold) than Formosa firethorn and field horsetail.

In addition, in terms of point of zero charge and zeta potential studies, it was determined that all biomaterials were negatively charged at their surface, agreeing with the molecular electrostatic potential surface data of this study. Regarding chlorophyll, this negative area was identified on the oxygen atoms of both the ketonic carbonyl and the ester carbonyl groups, positioned on the N4 ring. On the other hand, an important zone around the magnesium ion reveals that this site may be subject to a nucleophilic attack [21,35]. Therefore, for AFB_1_, the electrostatic potential is more reactive around the oxygen atoms localized on the D and E rings, accomplishing a nucleophilic attack. Furthermore, according to the atomic charge data in Appendix A for **chl a 1** and **chl a 2**, and their complexes, oxygen and nitrogen atoms possess negative charges, and the magnesium atom retains a positive charge, confirming the nucleophilic attack from oxygen atoms of AFB_1_ to **chl a 1** and to **chl a 2**. Complementarily, the HOMO–LUMO data show that the D and E rings are considered as the interaction sites with a cationic center (magnesium ion) [38], supporting the maximum hardness principle [40,41]. Consequently, the anticarcinogenic activity should be improved if a large quantity of chlorophyll is present in the biosorbents.

## 3. Materials and Methods

### 3.1. Quantum Chemical Calculations

Density functional theory [65,66] calculations were carried out using the Gaussian09 (Version 09, Gaussian, Inc., Wallingford CT, UK, 2013) [67] and Spartan’06 (Version 06, Wavefunction, Inc., Irvine CA, USA) [68] software. All the structures considered in the study, shown in Figure 1, were constructed using the graphical interface of Spartan ‘06 and Gaussview5 (Version 05, Gaussian, Inc., Wallingford CT, UK, 2013) [69] programs. Geometry optimizations and frequency analysis for the complexes and the individual fragments were carried out using the hybrid meta-generalized gradient approximation functional M06-2X [70]. It was efficacious to obtain configurations with all the real frequencies at this level. The choice of the M06-2X functional was based on the fact that it has been demonstrated as appropriate to perform modeling of these kinds of molecules [71,72], and the use of basis set 6-311G(d,p) [73]. Default convergence criteria were employed in each software package. The geometries of all the different configurations of **chl a**, and their complexes with AFB_1_, were fully optimized in the gas phase. The results confirm that the calculated geometry is at a minimum (all the normal modes are positive). The molecular electrostatic potential (MEP) map is habitually used as a predictive and interpretative tool in chemistry; creating a reactivity map displaying the molecular regions makes electrophilic and nucleophilic interactions more likely [74]. MEP maps can be obtained by mapping electrostatic potential onto the total electron density with a color code [75]. MEP contour maps provide a simple way to predict how different geometries could interact [76]. This property was determined using the DFT (M06-2X/6-311G(d,p) method. The HOMO–LUMO gap is a typical quantity used to describe the stability of certain molecules [77]. In this study, the corresponding calculations were based on a complete **chl a** model containing 135 atoms (**chl a**), as described in Figure 1. This model includes the whole structure of **chl a**, incorporating the phytol chain, not considered in several preceding studies on **chl a**.

The solvent effect was also calculated using the self-consistent reaction field (SCRF) method and the Tomasi’s polarizable continuum model (PCM), using water as a medium (dielectric value ε= 78.4) [78,79]. Natural bond orbital (NBO) was used for the electron natural population analysis in the Gaussian09 program. Natural population analysis (NPA) was used to compare differences rather than determining absolute atomic charges [80]. An atomic charge is not a physically obvious parameter; its value depends on the scheme by which the electron density of a molecule is partitioned. Additionally, a natural charge results from NPA in order to convert a wave function’s delocalized molecular orbitals into localized ones. This is performed to maintain all the information contained in the wave function [36]. The solvent effect and atomic charge were calculated at the same level of theory for all the molecules involved in the present study.

### 3.2. Docking Studies

The three-dimensional (3D) structure of **chl a 1**, **chl a 2**, and AFB_1_ were obtained as previously described. For this, **chl a 1** and **chl a 2** were employed as macromolecules, and AFB_1_ as a ligand. Docking studies were performed employing Autodock 4.2 [81]. A rectangular lattice (126 × 126 × 126) was superimposed on the entire macromolecule to achieve a blind docking procedure. All docking simulations were conducted using the hybrid Lamarckian genetic algorithm with an initial population of 100 randomly placed individuals and a maximum of 10^7^ energy evaluations. All other parameters were maintained at their default settings. The resulting docked orientations were clustered together, within a root-mean-square deviation (RMSD) of 0.5. Conformations with the lowest free energy binding (ΔG) and the highest frequency were selected employing Autodock tools [82]. The images were created using PyMol.

### 3.3. Molecular Dynamics Simulations

For the molecular dynamics (MD) simulations, **chl a 2** was submitted to MD employing the NAMD 2.6 software [83]. The parameters and topology for **chl a 2** were obtained employing the online SwissParam server (http:// www.swissparam.ch/ accessed on 6 April 2022). Hydrogen atoms were added with the psfgen command within the VMD program [84], and the structure was minimized using the steepest descent algorithm for 2000 steps using the CHARMM27 force field [85]. The resulting structure was immersed in water (10 Å TIP3 water model), and the charge was neutralized using 2 Cl^−^ ions. The particle mesh Ewald method [86] and periodic boundary conditions were applied to complete the electrostatic calculations. Additionally, Nose–Hoover Langevin piston pressure control was used, and maintained the temperature at 310 K [87]. The SHAKE method was used to provide an integration time step of 2 fs while keeping all bonds connected to the hydrogen atoms rigid [88]. The equilibration protocol involved 1500 minimization steps, followed by 30 ps of MD at 0 K for the water and ions while freezing the entire protein. Once the minimization of the entire system was achieved, the temperature was increased from 10 to 310 K over 30 ps to ensure that it would continue to modify its volume with 30 ps of NTP dynamics [88]. As a final step, the NTV dynamics continued for 50 ns. The trajectory of the system was stored every 1 ps, and the simulations were analyzed by capturing several snapshots every 1 ns. The snapshots and RMSD were obtained using the carma program [89]. The RMSD was analyzed to determine whether the protein had undergone a conformational change, because this value reflects the distance between pairs of the same atoms, represented by d, with respect to time. All the computational work was performed using pmemd cuda on an Intel Core i7–980x 3.33 Ghz Linux workstation with 12 Gb of RAM, 2 NVIDIA Geforce GTX530 video cards, and 1 NVIDIA Geforce GTX580 video card.

## 4. Conclusions

The M06-2X density functional led to the qualitative and quantitative description of chemical interactions among **chl a 1** and **chl a 2** with one and two AFB_1_ molecules. To our knowledge, **chl a 2** alludes to a novel conformation. The obtained results revealed that the interaction sites of **chl a 1** and **chl a 2** are explained using molecular electrostatic potential surface, molecular orbitals (HOMO and LUMO), and charge calculations; these properties were conveniently employed for the characterization and successful description of the preferred interaction sites, affording a well-founded explanation on behalf of the coordination of **chl a 1** and **chl a 2** with one and two AFB_1_ molecules. When the interaction between one AFB_1_ molecule and the chlorophyll molecule is β-oriented, the corresponding coordination product appeared more stable. However, when the interaction was between two AFB_1_ molecules and **chl a**, the **chl a 2**-E–2AFB_1_ interaction was the most stable. An energetic interaction difference with **chl a** was also found when comparing the oxygen’s carbonyl groups between the D and E rings. The acquired energy interactions between **chl a 1** and **chl a 2**, and the aflatoxin, considering water as the solvent, are lower than those shown in the gas phase. This is probably because **chl a 1** and **chl a 2** undergo a greater interaction with the medium than with AFB_1_. These findings were supported by results achieved from the docking studies, with the interaction between **chl a 2** and one AFB_1_ molecule being the most stable. This is because the folded conformation is preferred. In addition, regarding molecular dynamics simulations, **chl a 2** shows conformational changes along the entire simulation, meaning that it probably exists in folding and unfolding conformational equilibrium processes. Finally, considering all the calculated complexes (folded and unfolded α and β configurations and the involved carbonyl groups), the complexes with two AFB_1_ molecules were more stable than those with only one AFB_1_. Thus, it is important to highlight that, in general, biosorbents containing chlorophyll could improve AFB_1_ adsorption.

## Figures and Tables

**Figure 1 ijms-23-06068-f001:**
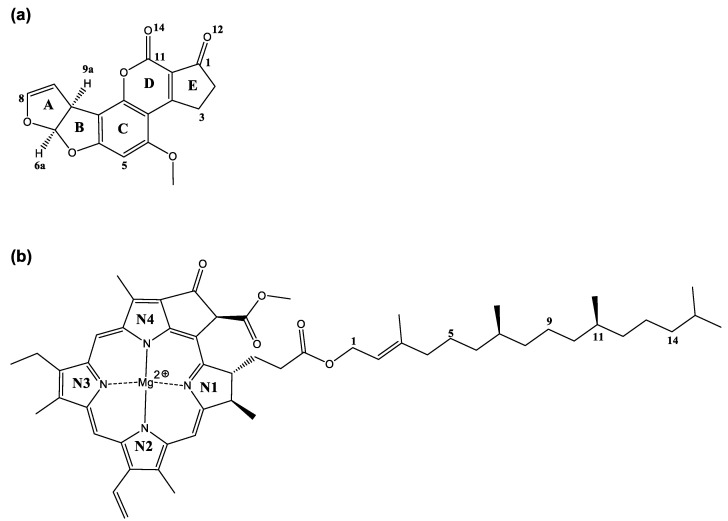
Structures of (**a**) aflatoxin B_1_ and (**b**) chlorophyll a (**chl a**).

**Figure 2 ijms-23-06068-f002:**
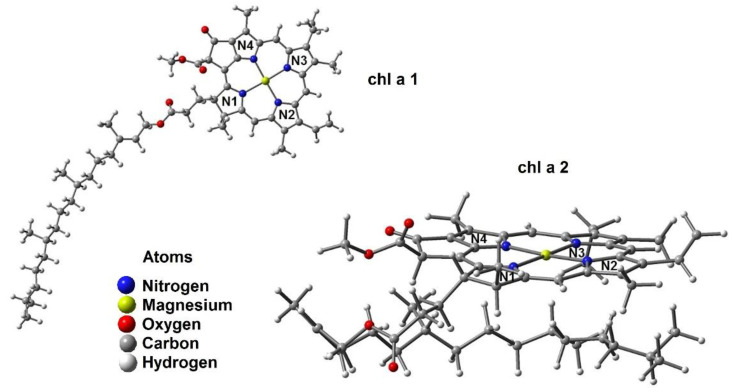
Optimized geometry of the more stable conformers: **chl a 1** and **chl a 2**.

**Figure 3 ijms-23-06068-f003:**
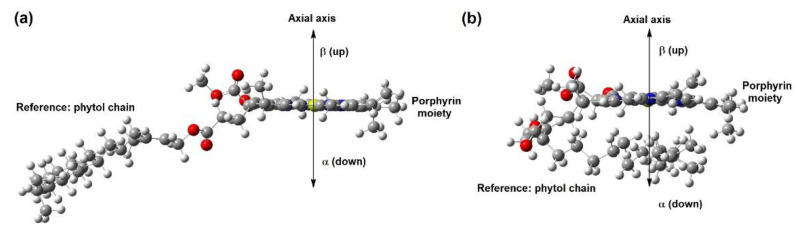
(**a**) Unfolded and (**b**) folded **chl a**.

**Figure 4 ijms-23-06068-f004:**
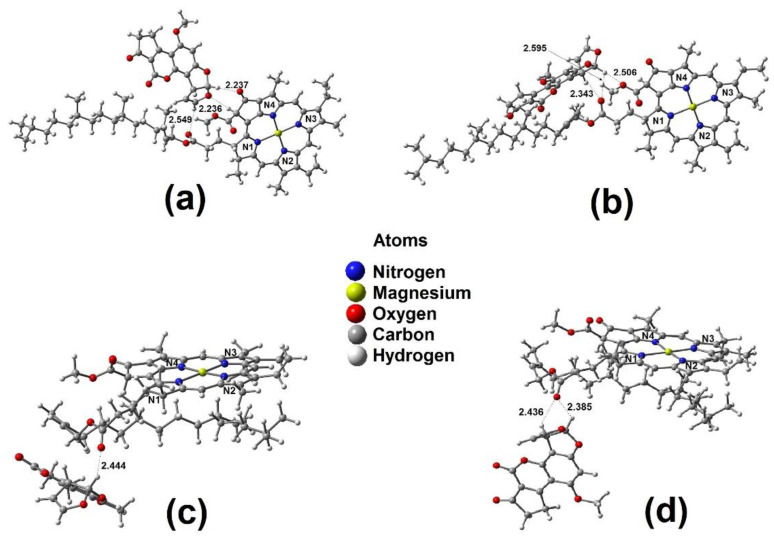
Optimized geometry of (**a**) **chl a 1**–AFB_1_a (three hydrogen bond interactions), (**b**) **chl a 1**–AFB_1_b (three hydrogen bond interactions); (**c**) **chl a 2**–AFB_1_c (one hydrogen bond interaction), and (**d**) **chl a 2**–AFB_1_d (two hydrogen bond interactions).

**Figure 5 ijms-23-06068-f005:**
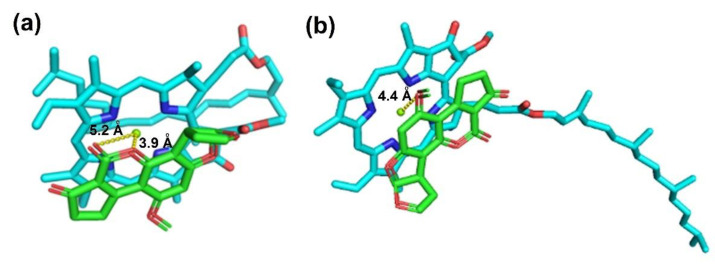
Interaction between chlorophyll and AFB_1_ obtained by docking studies; (**a**) **chl a 2**–AFB_1_ complex, (**b**) **chl a 1**–AFB_1_ complex.

**Figure 6 ijms-23-06068-f006:**
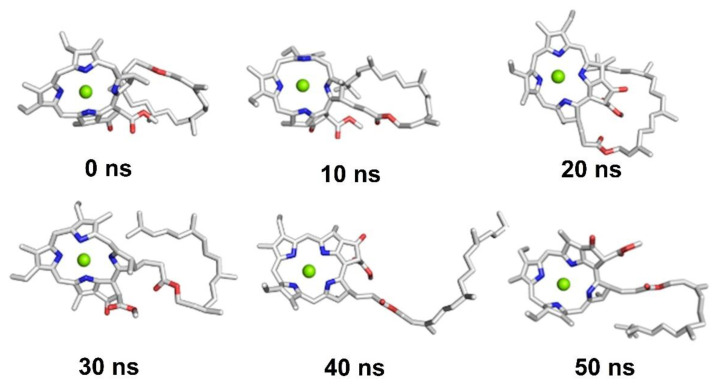
Snapshots for 50 ns of MD simulation of **chl a 2**.

**Table 1 ijms-23-06068-t001:** Bond distances, in Å, of optimized geometries.

Molecule	Mg–O1	Mg–N1	Mg–N2	Mg–N3	Mg–N4
**chl a 2**	-	2.131	2.013	2.061	2.001
**chl a 1**	-	2.133	2.015	2.063	2.002
**chl a 1**-α-D-AFB_1_	2.101	2.176	2.032	2.087	2.033
**chl a 1**-α-E-AFB_1_	2.106	2.183	2.037	2.076	2.036
**chl a 1**-β-D-AFB_1_	2.124	2.157	2.025	2.098	2.041
**chl a 1**-β-E-AFB_1_	2.111	2.160	2.023	2.092	2.050
**chl a 2**-α-D-AFB_1_	2.123	2.178	2.058	2.099	2.030
**chl a 2**-α-E-AFB_1_	2.080	2.170	2.045	2.088	2.020
**chl a 2**-β-E-AFB_1_	2.120	2.172	2.028	2.099	2.056
**chl a 2**-β-E-AFB_1_	2.119	2.151	2.057	2.101	2.016
**chl a 1**-D-2AFB_1_	2.230	2.135	2.043	2.084	2.024
	2.428 *				
**chl a 1**-E-2AFB_1_	2.228	2.113	2.025	2.114	2.030
	2.351 *				
**chl a 2**-D-2AFB_1_	2.285	2.128	2.054	2.084	2.019
	2.167 *				
**chl a 2**-E-2AFB_1_	2.251	2.149	2.027	2.092	2.028
	2.186 *				

E= oxygen atom of carbonyl group on ring E, D = oxygen atom of lactone group on ring D, * AFB_1_ molecule is α-located.

## Data Availability

The data reported in this study are available upon request to atlanta126@gmail.com or nicovain@yahoo.com.mx.

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
