# Peer review of "The Ability of Chlorophyll to Trap Carcinogen Aflatoxin B1: A Theoretical Approach"

_ijms, 2022, doi:10.3390/ijms23116068_

Round 1

Reviewer 1 Report

In the manuscript “The Ability of Chlorophyll to Trap Carcinogen Aflatoxin B1 in Foods, a Theoretical Approach”, authors have used density functional theory to study properties of chlorophyll and aflatoxin B1 complexes. I cannot recommend publication of this manuscript in the International Journal of Molecular sciences in its current form. However, this manuscript can be reconsidered after some major revisions.

1. Some sentences authors have added do not make any sense. For example-

"In this defined region, must be considered the interaction site"
"these results are in according with the molecular electrostatic" 

"Besides, in Table 5, are confined the atomic charges"

and many more.

Authors should proofread the manuscript more carefully.

2. I don't understand the section where the authors have discussed cutoff values and the HOMO-LUMO energy gap. Why is HOMOand LUMO energies changing when different cutoff values are used. If authors decide to keep this section in the manuscript, they need to provide more theoretical details about this feature they are using. At this point, it is not clear to me what this feature does.

I think the authors misunderstood my comment on the earlier review "What is the cutoff value of HOMO-LUMO gap for a complex to be stable?" I wanted to know if there is a value below which we can say a molecule is stable.

3. In section 2.12.1, H8, O7 and H6a are not labelled in figure 11.

Author Response

In the manuscript “The Ability of Chlorophyll to Trap Carcinogen Aflatoxin B1 in Foods, a Theoretical Approach”, authors have used density functional theory to study properties of chlorophyll and aflatoxin B1 complexes. I cannot recommend publication of this manuscript in the International Journal of Molecular sciences in its current form. However, this manuscript can be reconsidered after some major revisions.

Referee: 1. Some sentences authors have added do not make any sense. For example-
"In this defined region, must be considered the interaction site"
"these results are in according with the molecular electrostatic"
"Besides, in Table 5, are confined the atomic charges"
and many more.
Authors should proofread the manuscript more carefully.
Us: Done, with the English edition several mistakes were corrected,
consequently the manuscript was improved.

Referee: 2. I don't understand the section where the authors have discussed cutoff values and the HOMO-LUMO energy gap. Why is HOMO and LUMO energies changing when different cutoff values are used. If authors decide to keep this section in the manuscript, they need to provide more theoretical details about this feature they are using. At this point, it is not clear to me what this feature does.
I think the authors misunderstood my comment on the earlier review "What is the cutoff value of HOMO-LUMO gap for a complex to be stable?" I wanted to know if there is a value below which we can say a molecule is stable.
Us: Done, we sincerely apologize for the misunderstanding. Thus, kindly we
inform you about the changes.

A) The section related to the cutoff values was eliminated, in order to avoid
confusion.
B) To our knowledge, there is not any range-value reported in the literature. In
other words, the HOMO-LUMO GAP measures implicate relative values.

“In general, it has been established that if the HOMO-LUMO GAP value is larger,
the complex is most stable [1-4]. Thus, it is logical to think, that any value below the major value of GAP energy must be considered an unstable complex”.

In our case, the HOMO-LUMO GAP for chl a 1-AFB1 complex was 3.829 eV, and
for chl a 2-AFB1 complex was 3.841 eV, being this last the most stable complex.
Consequently, any value below of 3.841 eV must be considered an unstable
system. However, it is important to highlight that the HOMO-LUMO GAP
reported in this work was calculated between HOMO of AFB1 and LUMO of chl
a 1 and chl a 2, with 5.477 and 5.309 eV, respectively. This last interaction is
the most stable complex. The HOMO-LUMO GAP values are representative of
the fact that the formation of the complexes is taking place from the individual
molecules [5].

Please, see corresponding references below

References
1.‐ Üngӧrdϋ, A., Tezer, N. The solvent (water) and metal effects on HOMO‐LUMO gaps of guanine base pair: a computational study. J Mol Graph Model 2017, 74, 265‐272.
2.‐ Kumar Choudary, V.; Kumar Bhatt, A., Dash, D., Sharma, N. DFT calculations on molecular structures, HOMO‐LUMO study, reactivity descriptors and spectral analyses of newly synthesized diorganotin(IV) 2‐chloridophenylacetohydroxamate complexes. J Comput Chem 2019, 40, 2354‐2363.
3.‐ Soudani, S., Hajji, M., Xiao Mi, J., Jelsch, C., Lefebvre, F., Guerfel T., Ben Nasr C. Synthesis, structure and theoretical simulation of a zinc(II) coordination complex with 2,3‐pyridinedicarboxylate. J Mol Struct 2020, 1199, 1270158.
4.‐Rajaniverma, D., Jagadeeswara Rao, D., Rameeza Begum, Sk., Seetaramaiah, V., Ramakrishna Y. Optical, electrical, HOMO‐LUMO and first order hyperpolarizability studies on 2‐amino‐5‐bromopyridinium‐4‐hydroxybenzoate‐an organic single crystal for opto‐electronic applications. Mol Cryst Liq Crys 2021, 715, 69‐80.
5. An AM1 theoretical study on the effect of Zn2+ Lewis acid catalysis on the mechanism of the cycloaddition between 3‐phenyl‐1‐(2‐pyridyl)‐2‐propen‐1‐one and cyclopentadiene. C.N. Alves, A.B.F. da Silva, S. Marti, V. Moliner, M Oliva, J. Andrés, L. R. Domingo. Tetrahedron, 2002, 58, 2695‐2700.

Referee: 3. In section 2.12.1, H8, O7 and H6a are not labelled in figure 11.
Us: Done, a paragraph specifying where can be observed the atoms labeling
was included.

Referee: 4. Extensive editing of English language and style required
Us: Please see attached certificate. - We certify that the following article The Ability of Chlorophyll to Trap Carcinogen Aflatoxin B1, a Theoretical Approach Rene Miranda has undergone English language editing by MDPI. The text has been checked for correct use of grammar and common technical terms and edited to a level suitable for reporting research in a scholarly journal.

MDPI uses experienced, native English-speaking editors. Full details of the editing service can be found at https://www.mdpi.com/authors/english”.

Reviewer 2 Report

In the ijms-1707298 manuscript, the authors present a theoretical study of the coordination of one and two aflatoxin molecules with chlorophyll a. The topic is relevant and the computational level is adequate. However, the article is tedious to read (several details, Tables, and Figures should be moved to the Supporting Information). In addition, Section 2 should be summarized and the presentation of the results is confusing, and thus, a reorganization of the results can be useful for the potential reader of the article. Finally, taking into account that the manuscript is mainly focused on the interaction energies of feasible geometric interactions, I consider that this manuscript is more appropriate for a specialized journal.

Author Response

Referee: In the ijms-1707298 manuscript, the authors present a theoretical study of the coordination of one and two aflatoxin molecules with chlorophyll a. The topic is relevant and the computational level is adequate. However, the article is tedious to read (several details, Tables, and Figures should be moved to the Supporting Information). In addition, Section 2 should be summarized and the presentation of the results is confusing, and thus, a reorganization of the results can be useful for the potential reader of the article. Finally, taking into account that the manuscript is mainly focused on the interaction energies of feasible geometric interactions, I consider that this manuscript is more appropriate for a specialized journal.
Us: 1. Done, the work has been reorganized.
A) Several Figures and Tables were included in the Supporting Information,
and were reorganized.
1. Before Figure 12, currently Figure S1. M06-2X/6-311G(d,p) optimized geometries of AFB1. Atomic charges of selected atoms are in electron units.
2. Before Figure 3, currently Figure S2. Reactive sites attained in the electrostatic potential map. The total electron density isosurface mapped with the molecular electrostatic potential of AFB1, chl a 1, and chl a 2. The density = 0.0004 and isovalue = 0.02, at a level of calculation M06-2X/6-311G(d,p). The color scheme for the MEP is as follows: blue for electron-deficient, partially positive charge; light blue for the slightly electron-deficient region; yellow for a slightly electron-rich region and red for electron rich, partial negative charge [1].
3. Before Figure 4, currently Figure S3. HOMO (eV), LUMO (eV) and the inter-frontier molecular orbital energy gaps of the target molecules.
4. Before Figure 7, currently Figure S4. Optimized geometry: a) chl a 1-α-D-AFB1, b) chl a 1-α-E-AFB1, c) chl a 1-β-D-AFB1, and d) chl a 1-β-E-AFB1.
5. Before Figure 8, currently Figure S5. Optimized geometry: a) chl a 2-α-D-AFB1, b) chl a 2-α-E-AFB1, c) chl a 2-β-E-AFB1, and d) chl a 2-β-D-AFB1.
6. Before Figure 9, currently Figure S6. Optimized geometry of a) chl a 1-E-2AFB1, and b) chl a 1-D-2AFB1.
7. Before Figure 10, currently Figure S7. Optimized geometry of a) chl a 2-E-2AFB1, and b) chl a 2-D-2AFB1.
8. Before Figure 14, currently Figure S8. a) RMSD and b) radius of gyration of chl a 2, obtained by MD simulation.
9. Before Table 5, currently Table S1. Natural atomic charges (e–) for chl a 1, chl a
2, AFB1 and their complexes.
10. Before Table 3, currently Table S2. Interaction energy, in kcal/mol, of complexes chl a 1-AFB1 and chl a 2-AFB1 in gas phase.
11. Before Table 4, currently Table S3. Interaction energy, in kcal/mol, of complexes chl a 1-2AFB1 and chl a 2-2AFB1 in water as solvent.

B) With the reorganized work, section 2 was minimized. In this sense, the section was appropriately modified into nine subsections. The new version of the manuscript follows the next order:
2.1. DFT optimized structures
2.1.1. Determination of most stable conformer between chl a 1 and chl a 2
2.2. Structural and energetic parameters from optimized conformers
2.2.1. Atomic charges of chl a 1, chl a 2, AFB1, and their complexes
2.2.2. Bond distance (Å) of optimized geometries
2.2.3. Molecular electrostatic potential surface
2.2.4. Molecular orbitals: HOMO-LUMO for chl a 1, chl a 2 and AFB1
2.3. Geometry of unfolded and folded chlorophyll with an AFB1 molecule
2.3.1. Frontal view of unfolded and folded chlorophyll
2.3.2. Geometry of unfolded chlorophyll with an AFB1 molecule
2.3.3. Geometry of unfolded chlorophyll with two AFB1 molecules
2.3.4. Geometry of folded chlorophyll with an AFB1 molecule
2.3.5. Geometry of folded chlorophyll with two AFB1 molecules
2.4. Interaction energy of the complexes in gas phase
2.4.1. Considering the coupling of one AFB1 molecule with chl a 1 and chl a 2
2.4.2. Considering the coupling of two AFB1 molecules with chl a 1 and chl a 2
2.5. Interaction energy of the complexes in water as solvent
2.6. Weak hydrogen bond interactions between the ester functions of chl a 1 and 2 with AFB1
2.6.1. Chl a 1-AFB1a (three hydrogen bond interactions)
2.6.2. Chl a 1-AFB1b (three hydrogen bond interactions)
2.6.3. Chl a 2-AFB1c (one hydrogen bond interaction)
2.6.4. Chl a 2-AFB1d (two hydrogen bond interactions)
2.7. Docking studies for chl a 1, chl a 2-AFB1 complexes
2.8. Molecular dynamics (MD) simulations for chl a 2
2.9. Correlation between experimental and theoretical findings

In the same way, the Materials and Methods section was reorganized, with
three suitably modified subsections:
3.1. Quantum chemical calculations
3.2. Docking studies
3.3. Molecular dynamics simulations

C) Consequently, the References section was modified also. In this case, from
reference 36 to 89, the position were changed.

D) Related to the work, we decided to send it to the IJMS journal, specifically
to the Molecular Informatics section; sir, we sincerely are convinced that the
journal-section are appropriated; however, the Editor must have the right
decision.

Referee: 2. Moderate English changes required
Us: Please see attached certificate. - We certify that the following article The Ability of Chlorophyll to Trap Carcinogen Aflatoxin B1, a Theoretical Approach Rene Miranda has undergone English language editing by MDPI. The text has been checked for correct use of grammar and common technical terms and edited to a level suitable for reporting research in a scholarly journal. MDPI uses experienced, native English-speaking editors. Full details of the editing service can be found at https://www.mdpi.com/authors/english”.

Round 2

Reviewer 1 Report

Authors have addressed all the suggestions adequately. I recommend publication of this manuscript in its current form.

Reviewer 2 Report

The authors have mainly moved Figures and Tables to the SI and slightly modified the order of the presentation results; although the manuscript text remains similar. Consequently, if the Editor considers that the article fits the standards of the IJMS journal, it is fine for me. 

This manuscript is a resubmission of an earlier submission. The following is a list of the peer review reports and author responses from that submission.

Round 1

Reviewer 1 Report

In the manuscript “The Ability of Chlorophyll to Trap Carcinogen Aflatoxin B1 in Foods, a Theoretical Approach”, authors have used density functional theory to study properties of chlorophyll and aflatoxin B1 complexes. I cannot recommend publication of this manuscript in the International Journal of Molecular sciences in its current form. However, this manuscript can be reconsidered after some major revisions.

  • Authors have used very long sentences throughout the article. Those long sentences make it very difficult to understand the point the authors are making. Name of many of the sections are misleading- for example in the section “Considering only one interaction” authors did not consider one interaction but they have considered interaction between Chl and one AFB1 molecule. Sections names should not confuse the reader.

In the abstract, authors have mentioned M02-2X method instead of M06-2X.  

  • Throughout the manuscript authors have reported many data from their computation. However, they have not tried to make a connection of their finding with experimental findings. Authors should clearly state implications of their findings for the anticarcinogenic property of chlorophyll.
  • In the section about HOMO-LUMO gap of AFB1 and Chl, authors tried to justify stability of different Chl-AFB1 complexes using HOMO-LUMO gap of the individual molecules. What is the cutoff value of HOMO-LUMO gap for a complex to be stable? Can one make such an assumption just based on energy difference between HOMO and LUMO without taking into account orbital overlaps? Authors need to address this issues in this section.
  • Most problematic part of the manuscript is section 2.9 where authors have studied properties of different complexes in water. Authors have not provided any details about their calculations in water. Did they used any solvation model? If they used explicit solvation, how many water molecules did they consider? The sentence “The obtained results could involve an interaction between chl a-H2O.” indicates that authors may have used explicit water molecules. In that case how did they determine the position of the water molecules. Authors need to clarify the details of the solvation model they have used in this section.

Reviewer 2 Report

Notwithstanding the topic, the manuscript in the present form cannot be published because it lacks of important components. All the calculations have been performed in gas phase and this is a very high limitation. During the discussion appear some reference to water as solvent without having any information oh how the solvent effect has been considered.

energy values reported in hartrees in the text...

In general a full reorganization of work has been required. Methods are not well elucidated, are not clarified by expressions/chemical reaction interaction energies and so on

MD simulations can be required to better individuate the conformation of the examined molecules and in particular of the long hydrocarbon  skeleton.

Minor points

-in all the manuscript, including tables, energetical and geometrical values must be reported with dots and not with commas 

(-0.737 – 796e–) must be revised but exist many other similar typos.